# Association of Atherogenic Index of Plasma with Cardiometabolic Risk Factors and Markers in Lean 14-to-20-Year-Old Individuals: A Cross-Sectional Study

**DOI:** 10.3390/children10071144

**Published:** 2023-06-30

**Authors:** Katarína Šebeková, Radana Gurecká, Melinda Csongová, Ivana Koborová, Peter Celec

**Affiliations:** 1Institute of Molecular BioMedicine, Faculty of Medicine, Comenius University, 81108 Bratislava, Slovakia; radana.kollarova@gmail.com (R.G.); melinda.csongova@gmail.com (M.C.); koborova@gmail.com (I.K.); peter.celec@imbm.sk (P.C.); 2Institute of Medical Physics, Biophysics, Informatics and Telemedicine, Faculty of Medicine, Comenius University, 81108 Bratislava, Slovakia; 3Institute of Pathophysiology, Faculty of Medicine, Comenius University, 81108 Bratislava, Slovakia

**Keywords:** atherogenic index of plasma, lean, young adults, cardiometabolic risk factors, continuous metabolic syndrome score, blood count, sex hormones

## Abstract

Cardiometabolic risk factors at a young age pose a significant risk for developing atherosclerotic cardiovascular disease in adulthood. Atherogenic dyslipidemia is highly associated with obesity and metabolic syndrome already in young age. It remains unclear whether cardiometabolic risk factors associate with the atherogenic index of plasma (AIP = log (TAG/HDL-C) in lean subjects with low atherogenic risk. As both the AIP and markers of cardiometabolic risk are continuous variables, we expected their association to be linear before the manifestation of obesity and atherogenic dyslipidemia. We analyzed the prevalence of increased atherogenic risk (AIP ≥ 0.11) in 2012 lean 14-to-20-year-old subjects (55% females) and the trends of cardiometabolic risk factors across the quartiles (Q) of AIP in a subgroup of 1947 (56% females) subjects with low atherogenic risk (AIP < 0.11). The prevalence of AIP ≥ 0.11 reached 3.6% in females and 8.5% in males. HDL-C, non-HDL-C, triglycerides, and the continuous metabolic syndrome score showed a stepwise worsening across the AIP quartiles in both sexes. Measures of obesity and insulin resistance were worse in Q4 vs. Q1 groups, and leukocyte counts were higher in Q4 and Q3 vs. Q1. Females in Q4 presented with a higher C-reactive protein and lower adiponectin, estradiol, and testosterone levels. The multivariate regression model selected non-HDL-C, QUICKI, and erythrocyte counts as significant predictors of AIP in males; and non-HDL-C and C-reactive protein in females. A question arises whether the lean individuals on the upper edge of low atherogenic risk are prone to earlier manifestation of metabolic syndrome and shift to the higher AIP risk group.

## 1. Introduction

Atherosclerosis is a major underlying cause of various cardiovascular diseases (CVD). The higher the atherogenicity of plasma, the greater the risk of developing the conditions. An atherogenic index of plasma (AIP = log (triacylglycerols/high-density lipoprotein concentrations; TAG/HDL-C) [1,2] is an independent marker and predictor of CVD, and it might indicate a risk even when the components of AIP and the other atherogenic risk parameters appear normal [1,3]. AIP reflects the balance between protective and atherogenic lipoproteins and correlates with lipoprotein particle size and cholesterol esterification rates in apoB-lipoprotein-depleted plasma [1,2]. Based on AIP, individuals are classified into three risk groups: low (AIP < 0.11), intermediate (AIP: 0.11–0.21), and high atherogenic risk (AIP > 0.21) [2]. In the general population of adults, AIP is a powerful indicator of the presence of cardiometabolic risk factors [4], and a predictor of cardiometabolic diseases, major adverse cardiovascular events, and cardiovascular mortality [5,6,7].

Atherosclerotic cardiovascular disease risk factors presented during childhood and adolescence generally track into adulthood and increase the risk for cardiovascular events in later life. Atherogenic dyslipidemia is the most common dyslipidemia seen in children and adolescents and can be suspected based on the presence of risk factors, such as obesity or metabolic syndrome (MetS) [8,9]. Indeed, in children, adolescents, and young adults, AIP is associated with cardiometabolic risk factors, such as obesity, fatty liver, insulin resistance, and MetS [10,11,12,13,14]. However, except for Dağ et al.’s [13] study focusing on obese subjects, the mentioned investigations studied general populations. Using the harmonized criteria of MetS [15], Ostrihoňová et al. [16] showed that in Slovaks aged 10-to-17.9 years, the prevalence of central obesity is about 1.5-fold lower than that of hypertriacylglycerolemia and 4-fold lower than low HDL-C levels in males, and 2.3-fold and 8.4-fold, respectively, in females. This data [10,11,12,14,16] suggests that alterations in AIP may even precede obesity and abdominal adiposity, indicating that a proportion of young Slovaks presenting with increased AIP are lean. Data on the prevalence of increased AIP (>0.11) in lean subjects are missing. It also remains unclear whether and how cardiometabolic risk factors and markers associate with AIP in subjects at low atherogenic risk (AIP < 0.11) without obesity. Both the AIP and the factors and markers of cardiometabolic risk are continuous variables; thus, we expected their association to be linear before the manifestation of atherogenic dyslipidemia. Due to sex differences in the pathophysiology [17] and the presentation of MetS components [16], we anticipated sex differences in the prevalence of increased AIP in lean subjects, as well as in associations between AIP and different risk factors and markers. We analyzed data obtained from 2012 healthy, lean 14–20-year-old subjects to verify our hypothesis.

## 2. Materials and Methods

### 2.1. Study Design and Participants

During the academic year 2011/2012, students attending state secondary schools in the Bratislava Region willingly took part in the cross-sectional project called “Respect for Health” [18]. The primary objective of this study was to collect data that could be used to implement preventive health measures effectively. 

As previously mentioned [18], individuals with any acute or chronic illness and pregnant or lactating females were excluded. Anthropometric, blood chemistry, and hematology data were collected from a total of 2957 students aged 11 to 23 years. For the present analysis, only subjects between the ages of 14 and 20 years were included. General overweight/obesity was determined based on the International Obesity Task Force criteria for those aged 17 years or younger [19], while individuals aged 18 years or older with a BMI of 25.0 kg/m^2^ or higher were classified as overweight/obese. Subjects with a waist-to-height ratio (WHtR) of 0.5 or higher were considered centrally obese [20]. Following the exclusion of individuals with incomplete data for determining central obesity or general overweight/obesity, as well as those with CRP levels exceeding 10 mg/L, a total of 2659 subjects (51.7% females) remained for analysis. Data on the prevalence of increased AIP in the whole cohort (*n* = 2659) and overweight/obese subjects are given in the Appendix A. To investigate the associations between cardiometabolic risk markers and AIP in lean individuals, those classified as having central obesity or general overweight/obesity (referred to as overweight/obese hereafter) were excluded. This resulted in a final sample of 2012 lean subjects (54.7% females) without potential acute inflammation for analysis.

This study adhered to the principles outlined in the Helsinki Declaration. Approval for this study was obtained from the Ethics Committee of the Bratislava Self-Governing Region. Participation in the survey was voluntary, contingent upon obtaining written informed consent from participants aged 18 years or older and verbal assent from participants who were minors, with the consent of their legal representative.

### 2.2. Measurements

Trained personnel conducted anthropometric measurements according to established guidelines [18]. Height was measured using a portable extendable stadiometer, waist circumference was measured using a flexible tape, and body weight and total body fat percentage (TBF) were determined using digital scales (Omron BF510, Kyoto, Japan). Based on these measurements, body mass index (BMI) and waist-to-height ratio (WHtR) were calculated.

Blood pressure (B.P.) and heart rate (H.R.) were assessed on the dominant arm of each participant in a seated position after at least 5 min of relaxation. A digital monitor (Omron M-6 Comfort, Kyoto, Japan) was used for the measurements. The mean of the last two readings out of three was recorded for each participant.

After an overnight fasting period, venous blood samples were collected and transferred to the central laboratory for analysis. The following parameters were measured using standard laboratory methods (Advia 2400 analyzer, Siemens, Germany): serum fasting plasma glucose (FPG), triacylglycerols (TAG), high-density lipoprotein cholesterol (HDL-C), uric acid (U.A.), and creatinine. Immunoassay with direct chemiluminescence testing methodology (Advia Centaur XP Immunoassay System, Siemens, Germany) was used to quantify high-sensitivity C-reactive protein (CRP) and fasting plasma insulin (FPI). Blood counts were conducted using the Sysmex XE-2100 analyzer (Sysmex Corporation, Kobe, Japan) [18]. Total plasma L-homocysteine levels were measured using a fluorescence polarization immunoassay employing an Abbott IMX instrument (Abbott Diagnostics, Maidenhead, Berkshire, UK). Additionally, at the Institute of Molecular Biomedicine, serum adiponectin (R&D in Minneapolis, MN, USA) and total testosterone and estradiol (DRG Diagnostics, Marburg, Germany) levels were determined using ELISA methods according to the manufacturer’s instructions.

The formula of Dobiášová and Frölich [1] was used to calculate the atherogenic index of plasma. Based on the calculated values, subjects were categorized into two groups: low risk (with a value less than 0.11) and increased risk (with a value of 0.11 or higher). Non-HDL-C was determined by subtracting HDL-cholesterol from total cholesterol. Insulin sensitivity was assessed using the Quantitative Insulin Sensitivity Check Index (QUICKI) [21]. The estimated glomerular filtration rate (eGFR) was calculated using an equation that accounts for the entire age spectrum and incorporates Q-height extension [22]. 

The prevalence of elevated SBP (≥130 mm Hg), diastolic blood pressure (DBP) ≥85 mm Hg, FPG (≥5.6 mmol/L), non-HDL-C (≥3.8 mmol/L), TAG (>1.7 mmol/L), CRP (>3 mg/L), FPI (≥20 mlU/L) [23]; and low concentration of HDL-C (HDL-C < 1.03 in males and females aged <16 years, and <1.29 in females aged ≥16), and that of insulin resistance (QUICKI ≤ 319) [24] were determined. 

The continuous metabolic syndrome score (cMSS5), which serves as a proxy measure for cardiometabolic risk, was estimated using the formula developed by Soldatovic et al. [25]. Specifically, the cMSS5 score was calculated by summing the following components: WHtR/0.5 + FPG/5.6 + systolic blood pressure (SBP)/130 + TAG/1.7 +HDL-C/1.02 in all males and females aged ≤15 years, and /1.29 in females aged ≥16 years. Additionally, an alternative score (cMSS3) was computed by excluding variables related to the lipid profile. The cMSS3 score consisted of the following components: WHtR/0.5 + FPG/5.6 + SBP/130.

### 2.3. Sample Size Estimation

For multivariate regression analyses, the appropriate sample size is determined using either the participants-to-item ratio or a minimum required total sample size [26]. It is recommended to have a subject-to-variable ratio ranging between 10:1 and 30:1. The adequacy of the sample size can be assessed roughly based on the following scale: 50—very poor; 100—poor; 200—fair; 300—good; 500—very good; and ≥1000—excellent. In our study, we employed OPLS models with nine independent variables, and the number of participants ranged from 875 to 1100. Hence, our study meets even the most conservative requirement for the subject-to-variable ratio and aligns with the criteria of being classified as “very good” or even “excellent”.

### 2.4. Statistical Analysis

Variables that did not follow a normal distribution were transformed logarithmically. The two-way unpaired Student’s *t*-test was used to compare two sets of variables. The effect of AIP category, sex, and their interaction was analyzed using the general linear model. To capture potential nonlinear patterns of changes in cardiometabolic risk factors and markers among low-risk subjects across the AIP quartiles (Q), analysis of variance (ANOVA) was employed. The post hoc Bonferroni test was conducted to identify any significant differences. Pearson or Spearman correlation coefficients were calculated, and the two-sided Fisher r-to-z transformation was applied to determine the significance of differences between two correlation coefficients in two independent samples. Categorical data were compared using the chi-square test, and Yates’ correction was applied when necessary. To investigate the differences between males and females, separate evaluations were conducted for each sex. The data are presented as mean ± standard deviation (S.D.), geometric mean (with −1S.D. and +1S.D.) for back-transformed log data, and counts and percentages for categorical data. Statistical significance was determined using a *p*-value of less than 0.05. The statistical analyses were performed using SPSS software (v. 16 for Windows, SPSS, Chicago, IL, USA).

To identify independent variables that predict the atherogenic index of plasma (AIP), multivariate regression analysis was conducted using the orthogonal partial least square (OPLS) model. The analysis was performed using Simca software (v. 17, Sartorius Stedim Data Analytics AB, Umea, Sweden). Prior to the multivariate regression analysis, the independent variables were assessed for multicollinearity using the variance inflation factor (VIF). The following variables were included as independent determinants (predictors): WHtR, QUICKI, erythrocyte and leukocyte count, non-HDL-C, CRP, adiponectin, and sex hormone levels. Before modeling, variables with high skewness and low minimum/maximum ratio were log-transformed, and all data were mean-centered. Variables with a Variable Important for the Projection (VIP) value of ≥1.0 were considered significant predictors.

## 3. Results

### 3.1. Prevalence of Increased Atherogenic Risk

#### 3.1.1. Males

Out of 912 lean males, 37 (4.1%) displayed AIP ≥ 0.11: 20 (54.1%) presented with intermediate, and 17 with high atherogenic risk (AIP ≥ 0.21).

#### 3.1.2. Females

Out of 1100 lean females, 28 (2.5%) presented with increased atherogenic risk (AIP ≥ 0.11). Nine (32.1%) of them displayed intermediate, and nineteen were at high risk. 

#### 3.1.3. Between-Sex Comparison

The prevalence of AIP ≥ 0.11 was higher in the whole group of males and males presenting with overweight/obesity than in corresponding groups of females (*p*_Chi_ < 0.001, both, Appendix A). In lean subjects, neither the prevalence of increased atherogenic risk (*p*_Chi_ = 0.056) nor the proportion of subjects on intermediate vs. high risk (*p*_Chi_ = 0.078) differed significantly between sexes.

### 3.2. Characteristics of Lean Subjects

AIP ranged between −0.90 and 0.53 in males and −1.02 and 0.38 in females. Females displayed more favorable mean AIP than males, albeit they presented with higher levels of lipids and a higher prevalence of low HDL-C, elevated triacylglycerols, and non-HDL-C (Table 1). Significant between-sex differences were observed in all variables except for age and the prevalence of elevated DBP, fasting insulinemia, and insulin resistance.

### 3.3. Comparison of Lean Subjects with Low and Increased Atherogenic Risk

Subjects at increased risk had higher average values of standard body composition (Table 2). They presented with significantly higher fasting insulinemia, lower insulin sensitivity, less favorable lipid profile, higher CRP, homocysteine, erythrocyte counts, and lower sex hormone levels. Higher insulinemia, non-HDL-C, triacylglycerols, and CRP levels; lower insulin sensitivity and HDL-C concentrations associated with increased prevalence of these risk factors and markers beyond their threshold values. Cardiometabolic risk estimated as continuous MetS score was higher in both groups with increased AIP, even after excluding HDL-C and triacylglycerols from the equation (cMSS3). As sex or the sex*AIP interaction appeared significant in all analyses except for AIP and estradiol, further analyses were performed separately for males and females.

### 3.4. Relationship between AIP and Cardiovascular Risk Factors and Markers 

If all lean subjects were evaluated together, in both sexes proxy measures of obesity, concentrations of insulin, non-HDL-C, CRP, homocysteine, cMSS5, and leukocyte counts showed a direct—and QUICKI, a significant—inverse association with AIP (Table 3). AIP was associated significantly and positively with erythrocyte counts in males, while it negatively correlated with adiponectin and sex hormones in females. A comparison of correlation coefficients indicated that the association between AIP and CRP is more robust in females than males, while that with cMSS5 is more robust in males. However, except for nonHDL-C and cMSS, correlations were weak. 

The significance of the correlations in the whole cohort could have been caused due to the presence of subjects with increased atherogenic risk. To confirm or reject this assumption, correlations were recalculated after excluding subjects on increased risk (Table 3). This exclusion yielded only minor changes: weak associations between AIP and eGFR or CRP became insignificant in males. Correlations insignificant in both sexes (e.g., SBP, heart rate, FPG, uricemia) are given in Appendix A.

### 3.5. Cardiometabolic Risk Factors and Markers across the AIP Quartiles in Subjects on Low Risk

We evaluated the changes in risk factors and markers across AIP quartiles to capture potential nonlinear trends. 

#### 3.5.1. Males

As expected, based on employed categorization, AIP, HDL-C, triacylglycerols, and non-HDL-C showed significant trends across the AIP quartiles with significant differences between quartiles (Table 4). Similar trends and differences were observed for the continuous MetS score (cMSS5). However, after the exclusion of lipid profile variables, the trend in cMSS3 became insignificant (Q1: 2.63 ± 0.13, Q2: 2.64 ± 0.12, Q3: 2.64 ± 0.13, Q4: 2.66 ± 0.13; *p* = 0.129). Males in the highest AIP quartile presented with significantly higher BMI vs. Q1, waist circumference and WHtR vs. Q2, total body fat percentage vs. Q1 and Q2; and insulinemia, and lower QUICKI–both vs. Q1, Q2, and Q3. Erythrocyte and leukocyte counts were higher in Q4 and Q3 groups than in Q1. The prevalence of elevated fasting insulin, non-HDL-C, TAG, and CRP, and low HDL-C and QUICKI was significantly more frequent in males in the upper AIP quartile (Table 4). 

Variables displaying insignificant trends across the AIP quartiles in both sexes (age, SBP, DBP, heart rate, glycemia, uricemia, homocysteinemia, and eGFR; the prevalence of elevated blood pressure and FPG) are given for males in Appendix A.

#### 3.5.2. Females

Similar to males, AIP, variables characterizing lipid profile, and cMSS5 showed significant worsening across AIP quartiles and significant between-quartiles differences; and continuous MetS score calculated excluding HDL-C and TAG became insignificant (cMSS3: Q1: 2.40 ± 0.11, Q2: 2.40 ± 0.12, Q3: 2.47 ± 0.12, Q4: 2.49 ± 0.12; *p* = 0.184), (Table 5). 

Females in the upper AIP quartile displayed higher WHtR, BMI, adiponectin, and lower estradiol levels in comparison with those in Q1; higher total body fat percentage and lower testosterone levels vs. their peers in Q1 and Q2; and higher fasting insulinemia and lower QUICKI vs. the three lower quartiles. CRP levels and leukocyte counts were significantly higher in Q4 and Q3 females compared with those in the lowest quartile, and the Q4 group also showed higher levels vs. Q2. Significant trends in the prevalence of elevated non-HDL-C, TAG, and CRP, low HDL-C, and QUICKI were revealed (Table 5).

Variables displaying insignificant trends across the AIP quartiles in both sexes are given for females in Appendix A. 

### 3.6. Multivariate Regression of Cardiometabolic Risk Factors and Markers on the Atherogenic Index of Plasma

In all lean males, the OPLS regression model selected non-HDL-C, erythrocyte counts, QUICKI, and WHtR (VIP: 1.97–1.03) as significant predictors of AIP. In the subgroup of males on low risk, the WHtR became insignificant (Table 6). In both settings, the model poorly explained the variability of AIP (R^2^: 20%).

In females, the multivariate regression model selected non-HDL-C and CRP (VIP: 1.93 and 1.26 in all subjects, and 1.92 and 1.20 in females on low atherogenic risk) as significant predictors of AIP (Table 6). The models described 29% and 24%, respectively, in the variability of AIP (Table 6).

## 4. Discussion

We aimed to investigate the relationship between AIP and cardiometabolic risk factors and markers in lean young subjects. We confirmed our hypothesis that a proportion of young, lean individuals present with an increased atherogenic risk. The prevalence of subjects with AIP ≥ 0.11 was low and similar in males and females. Regardless of sex, lean subjects on low atherogenic risk displayed worsening trends across the AIP quartiles in proxy measures of obesity, insulin sensitivity, continuous MetS score, and leukocyte counts. Moreover, positive trends in erythrocyte counts were observed in males; while in females, CRP concentrations increased and sex hormone levels decreased across the quartiles. Except for the cMSS5, these trends were nonlinear; generally, significant worsening in Q4 vs. Q1 group was present.

In our subjects, the prevalence of AIP > 0.11 reached 3.6% in females and 8.5% in males. This prevalence is much lower than that reported for 5-to-19-year-old Chileans (54%) [10] or 18-to-22-year-old Mexicans (30%) [12]. Different prevalence was mirrored by differences in mean AIP values in these cohorts: negative in both sexes in our probands, varying around zero in Mexicans [12], and highly positive in the Chilean study [10]. Dissimilarities may stem from different prevalences of cardiometabolic risk factors since the studies in the general population of young subjects unequivocally report significant relationships between AIP and proxy measures of obesity, variables of lipid profile, and insulin sensitivity [10,11,12,14], but also, B.P. and uricemia [12,14]. Moreover, differences in lifestyle factors and genetic variations related to lipid concentrations might contribute [27].

In the general population of 40-year-old Slovaks, the prevalence of AIP > 0.11 reached 19% in females and 43% in males [28], and similar distributions were reported for both sexes of control groups of survivors of myocardial infarction [29]. The higher prevalence in studies on adults is in line with the fact that the degree of risk increases with age [30]. In Slovaks, the age-dependent rise probably reflects the increasing prevalence of hypertriacylglycerolemia: it is about two-fold higher in females and three-fold in males aged 35-to-45 years compared with their 10–18-year-old peers, while the prevalence of low HDL-C remains stable with increasing age [16].

To our knowledge, there is no data on the prevalence of increased AIP in lean individuals. Using a different approach (the Lipoprint system), Oravec et al. [31] concluded that the atherogenic lipoprotein profile might be present in about 6% of normolipidemic lean, healthy 14-year-olds. 

A review of 32 observational studies in adults indicated that all components of MetS, except hypertension, clearly associate with AIP, particularly in the presence of obesity [4]. Similar associations are described in the general population of children and adolescents or those with overweight/obesity [10,11,12,13,14,30]. We aimed to clarify whether associations between cardiometabolic risk factors and markers exist in low-risk subjects, even before the manifestation of overweight/obesity. 

Albeit AIP should be associated with higher TAG and lower HDL-C levels, the log transformation of their ratio does not provide information on whether their levels exceed the thresholds for MetS. These components of MetS showed the highest correlations with AIP in both sexes, even in subjects on low risk. As correlations may not be sensitive enough to detect variations occurring at the extremities of the distribution, we checked for their trends across AIP quartiles. While HDL-C levels declined linearly across the quartiles in both sexes, in line with the increasing prevalence of HDL-C below the cut-off for MetS, a linear rise in TAG levels across the quartiles was observed despite that all subjects with elevated TAG were grouped in the highest AIP quartile. These findings fit with a much higher prevalence of low HDL-C than high TAG in the Slovak population [32].

Proxy measures of central obesity associated with AIP in almost all studies [4,10,11,12,13,14,30]. Similar results were obtained for measures of general obesity, albeit BMI is not considered in the classification of MetS. We also observed positive correlations and increasing trends in measures of central obesity and general overweight/obesity (waist cf., WHtR, BMI, TBF percentage) across the AIP quartiles in lean subjects on low risk. However, these statistically significant results are of minor impact on clinical practice, as the differences in means between Q1 and Q4 are clinically negligible. In the multivariate regression model, WHtR (a proxy measure of central obesity with a single cut-off value for subjects of both sexes above five years) did not appear as a significant predictor of AIP in lean individuals with low risk. Quantifying visceral fat accumulation using magnetic resonance imaging or computer tomography could lead to conclusion on whether accumulation of visceral fat is associated with AIP before the manifestation of obesity in individuals at low risk. 

Associations between central obesity, insulin resistance, and inflammation are well-established [33,34]. Visceral fat accumulation is associated with hyperleptinemia and hypoadiponectinemia, resulting in hyperglycemia, hyperinsulinemia, hyperlipidemia, and inflammation. Low HDL-C or high TAG levels are independent predictors of insulin resistance [35,36]. In line with this data, even our lean subjects on low risk in Q4 maintained similar FPG with higher FPI, reflected by lower insulin sensitivity and lower QUICKI associated with its higher prevalence in Q4 subjects. However, in multivariate regression, QUICKI was a significant predictor of AIP only in males. As to inflammatory markers, leukocyte counts increased significantly (within the reference range) in the Q3 and Q4 groups of both sexes if compared with their Q1 counterparts. However, only CRP appeared as a significant predictor of AIP, and only in females. This finding aligns with our former observation that females present with elevated markers of inflammation before the manifestation of MetS, e.g., already when displaying one-to-two components of Mets [37]. 

Current knowledge shows that erythrocytes, generally considered passive gas carriers, are potent contributors to atherosclerotic plaque progression: erythrocytes colliding with the arterial wall induce local retention of their membranous lipids and hemolysis, releasing heme-Fe ^+ +^ with high toxicity for arterial endothelial and smooth muscle cells, promoting cell death [38,39]. Erythrocyte counts increased significantly (within their reference range) across the AIP quartiles only in males and were selected as significant predictors of AIP even in males on low risk. Ukrainian authors reported a significant direct relationship between erythrocyte counts and triacylglycerols in healthy adults, while the association with HDL-C was insignificant [40]. However, they did not investigate the associations of erythrocyte counts with atherogenic indices. In older adults without diabetes, overt cardiovascular and hematological diseases, insulin resistance, insulinemia, and triacylglycerols were associated directly, and concentrations of HDL-C inversely with erythrocyte counts [41]. Neither of these two studies tackled sex differences. Barbieri et al. [41] suggested that increased counts of erythrocytes represent a new aspect of insulin resistance syndrome, potentially contributing to an increased risk of developing cardiovascular diseases. This assumption was drawn from the data of in vitro studies, documenting how erythrocytes and their progenitors express insulin receptors [42,43]. Thus, insulin might exert proliferative effects at all stages of erythropoiesis. However, we neither observed sex differences in FPI levels across the AIP quartiles nor in the prevalence of elevated FPI; and insulinemia and QUICKI showed significant correlation with AIP in both sexes. We only might speculate that in lean young males, the association of erythrocyte counts with AIP is secondary, reflecting the sex-specific interrelation of AIP and yet unaltered insulin sensitivity. Adult males [44] and obese adolescent boys [45] are more insulin resistant than their female counterparts.

In adults, AIP has been shown to associate with lower eGFR and proposed as an indicator of the risk of developing renal impairment reflected by a decline in eGFR and an increase in microalbuminuria [46,47]. In our young lean probands on low risk, neither eGFR nor microalbuminuria estimated as albumin/creatinine ratio in spot urine (data not presented) associated significantly with AIP or showed a significant trend across the AIP quartiles. 

In adults of both sexes, low testosterone associates, independently of traditional risk factors, with increased all-cause and cardiovascular mortality [48,49]. In peri- and postmenopausal females, a decline in estrogens associated with accumulation of visceral fat, increase in TAG and decline in HDL-C levels, insulin resistance, and dysfunction of the vascular endothelium, eventually contributing to the increased risk of developing cardiometabolic diseases [50]. Our females with low risk displayed decreasing trends in testosterone and estradiol levels (within the reference range) across the AIP quartiles. As we neither have data on the menstrual cycle phase nor the use of contraceptives, interpretation of data is cumbersome. Aimed studies are needed to confirm a potential association of sex hormones with AIP in lean females at low risk.

The advantage of our study is a reasonably large cohort of young subjects of both sexes allowing for reliable multivariate analysis in lean subjects with low risk and a wide range of laboratory markers analyzed centrally. Limitations stem from the study’s cross-sectional nature: results are based on single measurements, allow only for comments on associations, and cannot be generalized to other populations. Additional limitations are mentioned in the discussion.

## 5. Conclusions

Here, we document that a proportion of lean 14-to-20-year-old subjects display an increased atherogenic risk, that the presence of central obesity or general overweight/obesity is not a prerequisite for the manifestation of known associations of cardiometabolic risk factors with the AIP, and that these associations exist even in lean subjects with low risk (AIP < 0.11). On the one hand, individuals in the highest AIP quartile present with less favorable cardiometabolic risk factors and markers compared with their Q1 peers, and continuous metabolic syndrome score (a measure of cardiometabolic risk) continuously increases across the AIP quartiles. Conversely, the analyzed factors and markers poorly explained the variation in AIP in lean subjects on low risk in multivariate regression. A question arises whether the lean individuals on the upper edge of low atherogenic risk are prone to earlier manifestation of MetS and shift to the intermediate or high-risk group. Interpretation of our findings from pathophysiological perspectives requires longitudinal studies.

## Figures and Tables

**Table 1 children-10-01144-t001:** Cohort characteristics.

	Males (*n* = 912)	Females (*n* = 1100)	*p*
Atherogenic index of plasma	−0.23 ± 0.20	−0.28 ± 0.20	**<0.001**
Age, years	17.2 ± 1.4	17.2 ± 1.4	0.644
Waist circumference, cm	75.3 ± 5.1	69.0 ± 5.0	**<0.001**
Waist/height	0.421 ± 0.027	0.416 ± 0.030	**<0.001**
Body mass index, kg/m^2^	21.1 ± 2.0	20.6 ± 2.0	**<0.001**
Total body fat, %	14.1 ± 4.6	28.1 ± 5.5	**<0.001**
Systolic blood pressure, mm Hg	120 ± 11	106 ± 9	**<0.001**
Diastolic blood pressure, mm Hg	72 ± 7	70 ± 7	**<0.001**
Heart rate, b/min	77 ± 13	81 ± 12	**<0.001**
Fasting plasma glucose, mmol/L	4.9 ± 0.4	4.7 ± 0.4	**<0.001**
Fasting plasma insulin, I.U./µL	8.4 (5.4; 13.1)	9.3 (6.0; 14.5)	**<0.001**
QUICKI	0.349 ± 0.025	0.347 ± 0.025	**0.025**
HDL-C, mmol/L	1.27 ± 0.23	1.55 ± 0.30	**<0.001**
nonHDL-C, mmol/L	2.46 ± 0.60	2.69 ± 0.68	**<0.001**
TAG, mmol/L	0.74 (0.51; 1.07)	0.79 (0.52; 1.19)	**<0.001**
Uric acid, µmol/L	344 ± 57	253 ± 49	**<0.001**
eGFR, ml/min/1.73 m^2^	111 ± 21	107 ± 16	**<0.001**
CRP, mg/L	0.4 (0.1, 1.1)	0.4 (0.1, 1.5)	**0.001**
Hcy, µmol/L	11.2 (7.7; 16.4)	9.4 (7.0; 12.7)	**<0.001**
Adiponectin, µg/mL	13.8 (7.0; 27.1)	18.9 (9.5; 37.7)	**<0.001**
cMSS5	1.86 ± 0.36	1.74 ± 0.37	**<0.001**
cMSS3	2.64 ± 0.13	2.48 ± 0.12	**0.044**
Erythrocytes, 10^12^/L	5.1 ± 0.3	4.5 ± 0.3	**<0.001**
Leukocytes, 10^9^/L	6.2 ± 1.4	6.7 ± 1.7	**<0.001**
Testosterone, nmol/L	19.6 (12.0; 31.8)	2.0 (1.3; 3.1)	**<0.001**
Estradiol, pmol/L	265 (183; 383)	329 (196; 554)	**<0.001**
Prevalence			*p* _Chi_
SBP ≥ 130 mm Hg, *n* (%)	170 (18.6)	14 (1.3)	**<0.001**
DBP ≥ 85 mm Hg, *n* (%)	42 (4.6)	35 (3.2)	0.103
Elevated SBP or DBP, *n* (%)	184 (20.2)	431 (3.9)	**<0.001**
Glucose ≥ 5.6 mmol/L, *n* (%)	59 (6.5)	19 (1.7)	**<0.001**
Insulin ≥ 20 mlU/L, *n* (%)	26 (2.9)	42 (3.8)	0.265
QUICKI ≤ 319, *n* (%)	84 (9.2)	120 (10.9)	0.235
HDL-C < 1.03 (males and females <16 years), <1.29 (females ≥ 16 years) mmol/L, *n* (%)	109 (12.0)	178 (16.2)	**0.007**
nonHDL-C ≥ 3.8 mmol/L, *n* (%)	24 (2.6)	72 (6.5)	**<0.001**
TAG > 1.7 mmol/L, *n* (%)	23 (2.5)	50 (4.5)	**0.016**
CRP > 3 mg/L, *n* (%)	47 (5.2)	82 (7.5)	**0.044**

Cf. circumference; WHtR waist to height ratio; BMI body mass index; TBF total body fat; SBP systolic blood pressure; DBP diastolic blood pressure; H.R. heart rate; FPG fasting plasma glucose; FPI fasting plasma insulin; QUICKI quantitative insulin sensitivity check index; HDL-C high-density lipoprotein cholesterol; TAG triacylglycerols; eGFR estimated glomerular filtration rate; CRP C-reactive protein; Hcy homocysteine; cMSS5 continuous metabolic syndrome score = WHtR/0.5 + FPG/5.6 + SBP/130 + TAG/1.7 +HDL-C/1.02 in all males and in females aged ≤ 15 years, and /1.29 in females aged ≥16 years; cMSS3 continuous metabolic syndrome score without lipids = WHtR/0.5 + FPG/5.6 + SBP/130; data are given as mean ± S.D. (normally distributed data) or as back-transformed log geometric mean (−1S.D.; +1S.D.) for data not fitting to normal distribution; data were compared using the two-sided unpaired Student’s *t*-test. *p* < 0.05 was considered significant (given in bold).

**Table 2 children-10-01144-t002:** Characteristics of lean subjects.

	Low Risk (AIP < 0.11)		Increased Risk (AIP ≥ 0.11)		*p*		
	Males	Females	Males	Females	Sex	AIPcat	S*AIP
*n* (%)	875 (43.5)	1072 (53.3)	37 (1.8)	28 (1.4)	−	−	−
AIP	−0.25 ± 0.17	−0.30 ± 0.18	0.23 ± 0.11	0.21 ± 0.07	0.114	**<0.001**	0.603
Waist cf., cm	75.2 ± 5.1	68.9 ± 5.0	78.2 ± 5.8	70.5 ± 6.5	**<0.001**	**<0.001**	**0.259**
WHtR	0.420 ± 0.027	0.416 ± 0.030	0.44 ± 0.03	0.42 ± 0.04	**0.009**	**<0.001**	0.151
BMI, kg/m^2^	21.1 ± 2.0	20.6 ± 2.0	22.0 ± 2.0	21.2 ± 2.2	**0.014**	**0.003**	0.621
TBF, %	14.1 ± 4.6	28.0 ± 5.5	15.4 ± 5.2	30.1 ± 6.3	**<0.001**	**0.008**	0.539
SBP, mm Hg	120 ± 11	106 ± 9	121 ± 11	107 ± 10	**<0.001**	0.682	0.902
DBP, mm Hg	72 ± 7	70 ± 7	73 ± 6	69 ± 8	**<0.001**	0.476	0.353
HR, b/min	77 ± 13	81 ± 12	75 ± 10	82 ± 11	**0.002**	0.589	0.314
FPG, mmol/L	4.9 ± 0.4	4.7 ± 0.4	4.9 ± 0.4	4.7 ± 0.4	**<0.001**	0.740	0.806
FPI, IU/µl	8.4 (5.4; 12.9)	9.3 (6.0; 14.4)	10.0 (5.4; 18.4)	12.7 (7.8; 20.9)	**0.003**	**<0.001**	0.220
QUICKI	0.350 ± 0.025	0.347 ± 0.025	0.342 ± 0.032	0.332 ± 0.024	**0.041**	**<0.001**	0.201
HDL-C, mmol/L	1.28 ± 0.22	1.55 ± 0.30	1.03 ± 0.20	1.35 ± 0.29	**<0.001**	**<0.001**	0.399
nonHDL-C, mmol/L	2.44 ± 0.58	2.66 ± 0.66	3.05 ± 0.86	3.70 ± 0.80	**<0.001**	**<0.001**	**0.019**
TAG, mmol/L	0.71 (0.51; 0.99)	0.77 (0.53; 1.13)	1.71 (1.34; 2.18)	1.65 (1.04; 2.61)	**0.001**	**<0.001**	0.124
Uric acid, µmol/L	344 ± 57	253 ± 49	351 ± 64	249 ± 50	**<0.001**	0.836	0.425
eGFR, ml/min/1.73 m^2^	111 ± 21	107 ± 16	104 ± 18	111 ± 15	0.474	0.502	**0.023**
CRP, mg/L	0.4 (0.1, 1.2)	0.4 (0.1, 1.6)	0.6 (0.2, 1.7)	1.3 (0.4, 4.4)	**0.001**	**<0.001**	0.084
Hcy, µmol/L	11.2 (7.7; 16.2)	9.4 (7.0; 12.7)	12.8 (8.3; 19.7)	10.1 (7.4; 13.8)	**<0.001**	**0.024**	0.385
Adiponectin, µg/mL	13.7 (7.0; 27.0)	18.9 (9.5; 37.7)	13.1 (6.8; 25.2)	16.7 (8.0; 35.3)	**0.001**	0.370	0.750
cMSS5	1.83 ± 0.31	1.71 ± 0.34	2.71 ± 0.34	2.71 ± 0.31	0.197	**<0.001**	0.150
cMSS3	2.64 ± 0.13	2.48 ± 0.12	2.68 ± 0.14	2.50 ± 0.15	**0.035**	**<0.001**	0.540
RBC, 10^12^/L	5.1 ± 0.3	4.5 ± 0.3	5.2 ± 0.3	4.6 ± 0.3	**<0.001**	**0.040**	0.446
WBC, 10^9^/L	6.2 ± 1.4	6.7 ± 1.7	6.6 ± 1.6	7.1 ± 2.0	**0.017**	0.057	0.876
TST, nmol/L	19.6 (12.1; 31.9)	2.0 (1.3; 3.1)	18.0 (10.8; 29.9)	1.7 (1.1; 2.7)	**<0.001**	**0.040**	0.446
Estradiol, pmol/L	265 (184; 383)	332 (197; 561)	237 (185; 305)	255 (171; 380)	0.266	**0.006**	0.031
Prevalence							
eSBP, *n* (%)	163 (18.6)	12 (1.1)	7 (18.9)	2 (7.1)	**<0.001**	0.368	0.413
eDBP, *n* (%)	40 (4.6)	34 (3.2)	2 (5.4)	1 (3.6)	0.508	0.801	0.929
eBP, *n* (%)	177 (20.2)	41 (3.8)	7 (18.9)	2 (7.1)	**<0.001**	0.797	0.553
eFPG, *n* (%)	56 (6.4)	17 (1.6)	3 (8.1)	2 (7.1)	0.236	0.137	0.430
eFPI, *n* (%)	23 (2.6)	38 (3.5)	3 (8.1)	4 (14.3)	0.122	**<0.001**	0.252
lQUICKI, *n* (%)	74 (8.4)	113 (10.5)	10 (27.0)	7 (25.3)	0.991	**<0.001**	0.591
lHDL-C, *n* (%)	90 (10.3)	166 (15.5)	19 (51.4)	12 (42.9)	0.707	**<0.001**	0.118
enon-HDL-C, *n* (%)	17 (1.9)	60 (5.6)	7 (18.9)	12 (42.9)	**<0.001**	**<0.001**	**<0.001**
eTAG, *n* (%)	3 (0.3)	26 (2.4)	20 (54.1)	24 (85.7)	**<0.001**	**<0.001**	**<0.001**
eCRP, *n* (%)	43 (4.9)	73 (6.8)	4 (10.8)	9 (32.1)	**<0.001**	**<0.001**	**0.002**

AIP atherogenic index of plasma; cf. circumference; AIPcat AIP category (low, increased risk); S*AIP interaction between sex and AIP categoty; WHtR waist to height ratio; BMI body mass index; TBF total body fat; SBP systolic blood pressure; DBP diastolic blood pressure; H.R. heart rate; FPG fasting plasma glucose; FPI fasting plasma insulin; QUICKI quantitative insulin sensitivity check index; HDL-C high-density lipoprotein cholesterol; TAG triacylglycerols; eGFR estimated glomerular filtration rate; CRP C-reactive protein; Hcy homocysteine; cMSS5 continuous metabolic syndrome score = WHtR/0.5 + FPG/5.6 + SBP/130 + TAG/1.7 +HDL-C/1.02 in all males and in females aged ≤ 15 years, and /1.29 in females aged ≥16 years; cMSS3 continuous metabolic syndrome score without lipids = WHtR/0.5 + FPG/5.6 + SBP/130; RBC erythrocytes; WBC leukocytes; TST testosterone; e elevated; l low; data are given as mean ± S.D. (normally distributed data) or as back-transformed log geometric mean (−1S.D.; +1S.D.) for data not fitting to normal distribution; continuous data were compared using the two-sided unpaired Student’s *t*-test, categorical data using the chi-square test; *p* < 0.05 was considered significant (given in bold).

**Table 3 children-10-01144-t003:** Correlations between cardiometabolic factors and markers and atherogenic index of plasma (AIP) in lean males and females.

	All					Low Risk				
	Males (*n* = 912)		Females (*n* = 1100)		*p* r to z	Males (*n* = 875)		Females (*n* = 1072)		*p* r to z
	r	*p*	r	*p*		r	*p*	r	*p*	
Waist cf.	0.133	**<0.001**	0.066	**0.028**	0.131	0.078	**0.010**	0.050	0.100	0.535
WHtR	0.135	**<0.001**	0.079	**0.009**	0.208	0.081	**0.016**	0.067	**0.028**	0.767
BMI	0.117	**<0.001**	0.100	**0.001**	0.704	0.086	**0.011**	0.086	**0.005**	1.000
TBF	0.130	**<0.001**	0.135	**<0.001**	0.912	0.117	**0.001**	0.119	**<0.001**	0.968
DBP	0.085	**0.010**	0.051	0.091	0.447	0.075	**0.026**	0.055	0.070	0.660
*FPI*	0.221	**<0.001**	0.207	**<0.001**	0.741	0.223	**<0.001**	0.190	**<0.001**	0.447
QUICKI	−0.216	**<0.001**	−0.199	**<0.001**	0.689	−0.213	**<0.001**	−0.176	**<0.001**	0.401
HDL-C	−0.580	**<0.001**	−0.405	**<0.001**	**<0.001**	−0.556	**<0.001**	−0.402	**<0.001**	**<0.001**
nonHDL-C	0.404	**<0.001**	0.444	**<0.001**	0.258	0.362	**<0.001**	0.399	**<0.001**	0.342
*TAG*	*0.915*	** *<0.001* **	*0.897*	** *<0.001* **	** *0.024* **	*0.904*	** *<0.001* **	*0.890*	** *<0.001* **	** *0.116* **
eGFR	−0.077	**0.019**	−0.013	0.672	0.153	−0.055	0.106	−0.027	0.376	0.542
*CRP*	*0.075*	** *0.024* **	*0.234*	** *<0.001* **	** *<0.001* **	*0.049*	*0.146*	*0.208*	** *<0.001* **	** *0.001* **
*Hcy*	*0.102*	** *0.002* **	*0.067*	** *0.026* **	*0.430*	*0.088*	** *0.010* **	*0.065*	** *0.034* **	*0.610*
*Adiponectin*	*−0.051*	*0.127*	*−0.147*	** *<0.001* **	** *0.031* **	*−0.048*	*0.165*	*−0.152*	** *<0.001* **	** *0.021* **
cMSS5	0.874	**<0.001**	0.826	**<0.001**	**<0.001**	0.837	**<0.001**	0.792	**<0.001**	**0.003**
cMSS3	0.098	**0.003**	0.043	0.152	0.219	0.078	**0.022**	0.035	0.254	0.342
RBC	0.189	**<0.001**	0.028	0.356	**<0.001**	0.190	**<0.001**	0.028	0.355	**0.001**
WBC	0.155	**<0.001**	0.158	**<0.001**	0.944	0.152	**<0.001**	0.158	**<0.001**	0.897
*TST*	−0.034	0.389	−0.154	**<0.001**	**0.007**	−0.019	0.644	−0.195	**<0.001**	**<0.001**
*E2*	−0.054	0.173	−0.175	**<0.001**	**0.006**	−0.048	0.501	−0.149	**<0.001**	**0.025**

cf. circumference; WHtR waist to height ratio; BMI body mass index; TBF total body fat; DBP diastolic blood pressure; H.R. heart rate; FPI fasting plasma insulin; QUICKI quantitative insulin sensitivity check index; HDL-C high-density lipoprotein cholesterol; TAG triacylglycerols; eGFR estimated glomerular filtration rate; CRP C-reactive protein; Hcy homocysteine; cMSS5 continuous metabolic syndrome score = WHtR/0.5 + FPG/5.6 + SBP/130 + TAG/1.7 +HDL-C/1.02 in all males and in females aged ≤15 years, and /1.29 in females aged ≥16 years; cMSS3 continuous metabolic syndrome score without lipids = WHtR/0.5 + FPG/5.6 + SBP/130; RBC erythrocytes; WBC leukocytes; TST testosterone; E2 estradiol; Pearson correlation coefficients were calculated for normally distributed data; Spearman’s Rho (given in Italics) was used to evaluate skewed data; Fisher’s r to z transformation was used to assess the significance of the difference between two correlation coefficients in two independent samples (males vs. females); significant correlations are given in bold.

**Table 4 children-10-01144-t004:** Data according to the quartiles of the atherogenic index of plasma in males.

	Q1 (*n* = 218) (−0.90, −0.37]	Q2 (*n* = 219) (−0.37, −0.25]	Q3 (*n* = 219) (−0.25, −0.12]	Q4 (*n* = 219) (−0.12, 0.11]	*p*
AIP	−0.47 ± 0.08	−0.31 ± 0.04 ***	−0.19 ± 0.04 ***^,+++^	−0.03 ± 0.06 ***^,+++,###^	**<0.001**
Waist cf., cm	74.9 ± 4.8	74.7 ± 5.1	75.2 ± 5.3	76.0 ± 5.1 ^+^	**0.027**
WHtR	0.420 ± 0.025	0.416 ± 0.027	0.420 ± 0.027	0.426 ± 0.027 ^++^	**0.002**
BMI, kg/m^2^	20.9 ± 1.9	21.0 ± 1.9	21.0 ± 2.1	21.4 ± 1.9 *	**0.030**
TBF, %	13.4 ± 4.4	13.8 ± 4.3	14.1 ± 4.7	15.0 ± 4.7 **^,+^	**0.003**
FPI, IU/µl	7.5 (4.8; 11.5)	8.1 (5.5; 11.9)	8.2 (5.6; 12.2)	10.4 (6.5; 16.7) ***^,+++,###^	**<0.001**
QUICKI	0.356 ± 0.026	0.350 ± 0.022	0.350 ± 0.023	0.341 ± 0.025 ***^,++,##^	**<0.001**
HDL-C, mmol/L	1.46 ± 0.22	1.32 ± 0.19 ***	1.22 ± 0.16 ***^,+++^	1.14 ± 0.17 ***^,+++,###^	**<0.001**
nonHDL-C, mmol/L	2.18 ± 0.50	2.34 ± 0.54 *	2.48 ± 0.53 ***^,+^	2.74 ± 0.59 ***^,+++,###^	**<0.001**
TAG, mmol/L	0.48 (0.39; 0.60)	0.64 (0.55; 0.74) ***	0.78 (0.67; 0.90) ***^,+++^	1.06 (0.89; 1.26) ***^,+++,###^	**<0.001**
CRP, mg/L	0.3 (0.1, 1.0)	0.3 (0.1, 1.0)	0.4 (0.1, 1.2)	0.4 (0.1, 1.35)	0.186
Adiponectin, µg/mL	13.4 (6.9; 26.2)	15.2 (7.5; 30.7)	13.5 (7.0; 25.7)	13.1 (6.6; 26.1)	0.098
cMSS5	1.49 ± 0.23	1.73 ± 0.18 ***	1.91 ± 0.16 ***^,+++^	2.17 ± 0.18 ***^,+++,##^	**<0.001**
Erythrocytes, 10^12^/L	5.0 ± 0.3	5.1 ± 0.3	5.1 ± 0.3 **	5.2 ± 0.3 ***	**<0.001**
Leukocytes, 10^9^/L	5.9 ± 1.5	6.2 ± 1.3	6.3 ± 1.4 *	6.5 ± 1.5 ***	**0.001**
TST	19.6 (12.3; 31.3)	20.3 (13.6; 30.3)	19.7 (11.3; 34.4)	19.0 (11.6; 31.3)	0.694
Estradiol	265 (187; 376)	273 (199; 374)	256 (169; 389)	267 (191; 374)	0.510
Prevalence					*p* _chi_
Insulin ≥20 mlU/L, *n* (%)	6 (2.8)	2 (0.9)	4 (1.8)	11 (5.0)	**0.046**
QUICKI ≤319, *n* (%)	13 (6.0)	12 (5.5)	14 (6.4)	35 (16.0)	**<0.001**
HDL-C <1.03 (M), *n* (%)	2 (0.9)	12 (5.5)	19 (8.7)	57 (26.0)	**<0.001**
nonHDL-C ≥ 3.8 mmol/L, *n* (%)	2 (0.9)	1 (0.5)	4 (1.8)	10 (4.6)	**0.009**
TAG >1.7 mmol/L, *n* (%)	0	0	0	3 (1.4)	**0.029**
CRP >3 mg/L, *n* (%)	5 (2.3)	7 (3.2)	12 (5.5)	20 (9.1)	**0.005**

Q quartile; Cf. circumference; WHtR waist to height ratio; BMI body mass index; TBF total body fat; FPI fasting plasma insulin; QUICKI quantitative insulin sensitivity check index; HDL-C high-density lipoprotein cholesterol; TAG triacylglycerols; CRP C-reactive protein; cMSS5 continuous metabolic syndrome score = WHtR/0.5 + FPG/5.6 + SBP/130 + TAG/1.7 + HDL-C/1.02; TST testosterone; Chi chi-square; data are given as mean ± S.D. (normally distributed data) or as back-transformed log geometric mean (−1S.D.; +1S.D.) for data not fitting to normal distribution; data were compared using the ANOVA test with post hoc Bonferroni correction; * *p* < 0.05 vs. Q1; ** *p* < 0.01 vs. Q1; *** *p* < 0.001 vs. Q1; ^+^
*p* < 0.05 vs. Q2; ^++^
*p* < 0.01 vs. Q2; ^+++^
*p* < 0.001 vs. Q2; ^##^
*p* < 0.01 vs. Q3; ^###^
*p* < 0.001 vs. Q3. *p* < 0.05 was considered significant (given in bold).

**Table 5 children-10-01144-t005:** Data according to the quartiles of the atherogenic index of plasma in females.

	Q1 (*n* = 268) (−1.02, −0.42]	Q2 (*n* = 268) (−0.42, −0.29]	Q3 (*n* = 268) (−0.29, −0.17]	Q4 (*n* = 268) (−0.17, −0.11]	*p*
AIP	−0.53 ± 0.09	−0.38 ± 0.04 ***	−0.23 ± 0.03 ***^,+++^	−0.07 ± 0.07 ***^,+++,###^	**<** **0.001**
WHtR	0.414 ± 0.032	0.414 ± 0.030	0.416 ± 0.030	0.421 ± 0.030 *	**0.024**
BMI, kg/m^2^	20.3 ± 2.0	20.6 ± 2.0	20.7 ± 2.0	20.8 ± 2.0 *	**0.017**
TBF, %	27.2 ± 5.5	27.5 ± 5.9	28.3 ± 5.4	29.0 ± 5.1 **^,++^	**<** **0.001**
FPI, IU/µl	8.4 (5.4; 13.3)	8.8 (5.7; 13.6)	9.1 (6.1; 13.5)	10.8 (7.0; 16.5) ***^,+++,###^	**<** **0.001**
QUICKI	0.352 ± 0.027	0.350 ± 0.024	0.348 ± 0.023	0.339 ± 0.023 ***^,+++,###^	**<** **0.001**
HDL-C, mmol/L	1.72 ± 0.29	1.58 ± 0.26 ***	1.49 ± 0.26 ***^,++^	1.42 ± 0.29 ***^,+++,##^	**<** **0.001**
nonHDL-C, mmol/L	2.38 ± 0.57	2.55 ± 0.57 *	2.70 ± 0.61 ***^,+^	3.03 ± 0.68 ***^,+++,###^	**<** **0.001**
TAG, mmol/L	0.50 (0.40; 0.63)	0.68 (0.57; 0.82) ***	0.86 (0.71; 1.04) ***^,+++^	1.19 (0.93; 1.51) ***^,+++,###^	**<** **0.001**
CRP, mg/L	0.3 (0.1, 0.9)	0.4 (0.1, 1.3)	0.5 (0.1, 1.5) **	0.6 (0.2, 2.2) ***^,+++^	**<** **0.001**
Adiponectin, µg/mL	21.8 (11.3; 42.2)	19.1 (9.6; 38.0)	19.0 (9.6; 37.3)	16.4 (8.1; 33.1) ***	**<** **0.001**
cMSS5	1.39 ± 0.25	1.61 ± 0.22 ***	1.78 ± 0.20 ***^,+++^	2.08 ± 0.22 ***^,+++,##^	**<** **0.001**
Erythrocytes, 10^12^/L	4.5 ± 0.3	4.5 ± 0.3	4.5 ± 0.3	4.5 ± 0.3	0.940
Leukocytes, 10^9^/L	6.4 ± 1.6	6.5 ± 1.7	6.8 ± 1.6 *	7.1 ± 1.9 ***^,++^	**<** **0.001**
TST	2.2 (1.5; 3.3)	2.1 (1.3; 3.4)	2.0 (1.4; 3.0)	1.9 (1.2; 2.8) **^,+^	**0.003**
Estradiol	359 (220; 588)	338 (200; 572)	333 (196; 566)	301 (177; 514) **	**0.010**
Prevalence					*p* _chi_
Insulin ≥ 20 mlU/L, *n* (%)	8 (3.0)	12 (4.5)	5 (1.9)	13 (4.9)	0.215
QUICKI ≤319, *n* (%)	22 (8.2)	28 (10.4)	21 (7.8)	42 (15.7)	**0.011**
HDL-C (<1.03 aged ≤15, <1.29 aged >15 years) mmol/L, *n* (%)	8 (3.0)	23 (8.6)	45 (16.8)	90 (33.6)	**<** **0.001**
nonHDL-C ≥3.8 mmol/L, *n* (%)	6 (2.2)	5 (1.9)	12 (4.5)	37 (13.8)	**<** **0.001**
TAG >1.7 mmol/L, *n* (%)	0	0	0	26 (9.7)	**<** **0.001**
CRP >3 mg/L, *n* (%)	9 (3.4)	16 (6.0)	18 (6.7)	30 (11.2)	**0.004**

Q quartile; Cf. circumference; WHtR waist to height ratio; BMI body mass index; TBF total body fat; FPI fasting plasma insulin; QUICKI quantitative insulin sensitivity check index; HDL-C high-density lipoprotein cholesterol; TAG triacylglycerols; CRP C-reactive protein; cMSS5 continuous metabolic syndrome score = WHtR/0.5 + FPG/5.6 + SBP/130 + TAG/1.7 + HDL-C/1.02; TST testosterone; chi-square test; data are given as mean ± S.D. (normally distributed data) or as back-transformed log geometric mean (−1S.D.; +1S.D.) for data not fitting to normal distribution; data were compared using the ANOVA test with post hoc Bonferroni correction; * *p* < 0.05 vs. Q1; ** *p* < 0.01 vs. Q1; *** *p* < 0.001 vs. Q1; ^+^
*p* < 0.05 vs. Q2; ^++^
*p* < 0.01 vs. Q2; ^+++^
*p* < 0.001 vs. Q2; ^##^
*p* < 0.01 vs. Q3; ^###^
*p* < 0.001 vs. Q3. *p* < 0.05 was considered significant (given in bold).

**Table 6 children-10-01144-t006:** Multivariate regression of selected cardiovascular risk factors and markers (independent variables) on the atherogenic index of plasma (dependent variables), using the orthogonal projections to latent structures model in lean subjects.

	Males		Females	
	VIP			
	All	Low Risk	All	Low Risk
Non-high-density lipoprotein cholesterol	**1.97**	**1.90**	**1.93**	**1.92**
Erythrocyte count	**1.15**	**1.12**	0.28	0.31
QUICKI	**1.05**	**1.15**	0.89	0.87
Waist/height	**1.03**	0.98	0.82	0.87
Leukocyte count	0.94	0.87	0.69	0.78
C-reactive protein	0.66	0.71	**1.26**	**1.20**
Testosterone	0.51	0.56	0.75	0.72
Estradiol	0.18	0.31	0.79	0.67
Adiponectin	0.05	0.42	0.71	0.82
R^2^	0.20	0.20	0.29	0.24

QUICKI quantitative insulin sensitivity check index; Variables with a variable of importance for the projection values ≥1.00 were considered important (significant) contributors (provided in bold).

## Data Availability

The data that support the findings of this study are available from the corresponding author upon reasonable request.

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
