# Peer review of "Association of Atherogenic Index of Plasma with Cardiometabolic Risk Factors and Markers in Lean 14-to-20-Year-Old Individuals: A Cross-Sectional Study"

_children, 2023, doi:10.3390/children10071144_

Round 1

Reviewer 1 Report

The authors present an original study entitled “Association of Atherogenic Index of Plasma with Cardiometabolic Risk Factors and Markers in Lean 14-to-20-Year-old Individuals: A Cross-sectional Study”.

The results of the presented study are interesting and novel. The results are of high clinical relevance as they provide evidence of the potential utility of AIP measuring in routine clinical examinations of the general population, including children and adolescents. Sufficient sample size and appropriate statistical analysis methods determine the validity of the findings.

I have a few minor points to address:

1.    “Currently, an atherogenic index of plasma (AIP = log (triacylglycerols/high-density lipoprotein concentrations; TAG/HDL-C) is considered the most accurate independent marker and predictor of CVD…”

The most accurate of a plethora of biomarkers? I would recommend using gentler terms.

2.    “…insulin (FPI)…and high-sensitivity C-reactive protein (CRP) employing standard laboratory methods…”.

Please provide specific methods and kits used (as you did for L-homocysteine).

3.    Table 4 requires an explanation of the symbols used (***, +++, ###).

4.    In the Discussion section, the authors briefly discussed the observed changes in erythrocyte counts as a function of AIP values (“Erythrocyte counts increased significantly (within their reference range) across the AIP quartiles only in males and were selected as significant predictors of AIP even in males on low risk”).

It would be good if the authors could provide a possible mechanistic explanation for this particular finding in men.

Author Response

The authors present an original study entitled “Association of Atherogenic Index of Plasma with Cardiometabolic Risk Factors and Markers in Lean 14-to-20-Year-old Individuals: A Cross-sectional Study”.

The results of the presented study are interesting and novel. The results are of high clinical relevance as they provide evidence of the potential utility of AIP measuring in routine clinical examinations of the general population, including children and adolescents. Sufficient sample size and appropriate statistical analysis methods determine the validity of the findings.

Response: We thank the reviewer for her/his time and valuable and insightful comments that led to possible improvements in the current version of our paper. We tried our best to address each comment, and the implemented changes are given in red.

I have a few minor points to address:

  1. “Currently, an atherogenic index of plasma (AIP = log (triacylglycerols/high-density lipoprotein concentrations; TAG/HDL-C) is considered the most accurate independent marker and predictor of CVD…”

The most accurate of a plethora of biomarkers? I would recommend using gentler terms.

Response: We agree with reviewer´s remark, and the sentence has been reformulated in the revised paper (lines 37-40).

  1. “…insulin (FPI)…and high-sensitivity C-reactive protein (CRP) employing standard laboratory methods…”.

Please provide specific methods and kits used (as you did for L-homocysteine).

Response: As suggested, in the revised paper we included the description of the method used to determine hsCRP and insulin levels (immunoassay, Advia Centaur XP analyzer), lines 112-114.

  1. Table 4 requires an explanation of the symbols used (***, +++, ###).

Response: We thank the opponent for alerting us to this omission. The explanation of the symbols used has been added in the revised paper (lines 288-290; 310-312).

  1. In the Discussion section, the authors briefly discussed the observed changes in erythrocyte counts as a function of AIP values (“Erythrocyte counts increased significantly (within their reference range) across the AIP quartiles only in males and were selected as significant predictors of AIP even in males on low risk”).

It would be good if the authors could provide a possible mechanistic explanation for this particular finding in men.

Response: As suggested by the reviewer, we tried to provide an explanation for this unexpected finding (lines 410-427).

Reviewer 2 Report

I was happy to evaluate this article by Sebekova et al.

This is an interesting, well documented and well written study with interesting discussions and important conclusions. Some concerns, however, are listed below.

I am a bit confused as to why some overweight or obese patients seem to be included in the study group. The title refers to lean patients only. Please define the inclusion/exclusion criteria better (especially in terms of overweight and obesity), so that there is no misunderstanding.

The pariticipants paragraph states: “To study the associations of cardiometabolic risk markers with AIP in lean subjects, those with central obesity and general overweight/obesity (from now on referred to as overweight/obese) were excluded, leaving 2,012 lean subjects (54.7% females) without potential acute inflammation for analysis.

Yet the results paragraph discusses the incidence of overweight and obese patients. This induces confusion.

I do not understand the following sentence: To affirm that correlations were driven solely by the small subgroup on increased atherogenic risk, correlations were recalculated after excluding subjects on increased risk. Please rephrase.

Reference 45 is incorrect.

Author Response

I was happy to evaluate this article by Sebekova et al.

This is an interesting, well documented and well written study with interesting discussions and important conclusions. Some concerns, however, are listed below.

Response: We thank the reviewer for giving us the opportunity to submit a revised draft of our manuscript. We appreciate the time and effort spent revising our paper. We incorporated changes to reflect the suggestions and have highlighted the changes within the manuscript in red.

I am a bit confused as to why some overweight or obese patients seem to be included in the study group. The title refers to lean patients only. Please define the inclusion/exclusion criteria better (especially in terms of overweight and obesity), so that there is no misunderstanding.

The pariticipants paragraph states: “To study the associations of cardiometabolic risk markers with AIP in lean subjects, those with central obesity and general overweight/obesity (from now on referred to as overweight/obese) were excluded, leaving 2,012 lean subjects (54.7% females) without potential acute inflammation for analysis.

Yet the results paragraph discusses the incidence of overweight and obese patients. This induces confusion.

Response: We agree quoting the overall prevalence of increased AIP and the prevalence in overweight/obese subjects within the results might be confusing. However, be supposed that this data might be of interest to some readers to get an idea of the overall population from which the lean subjects were selected. Thus, we moved the data on the prevalence of increased AIP in all overweight/obese subjects to supplementary material, as indicated in the revised paper in lines 86-88. In the revised paper, we only mention the characteristics of lean subjects. Thus, the evaluated cohort meets the inclusion/exclusion criteria listed in the Methods section.

I do not understand the following sentence: “To affirm that correlations were driven solely by the small subgroup on increased atherogenic risk, correlations were recalculated after excluding subjects on increased risk”. Please rephrase.

Response: As suggested by the reviewer, the sentence has been reformulated (lines 244-246).

Reference 45 is incorrect.

Response: We apologize for this mistake. As suggested, the statement not fitting references has been removed from the list.